# A theory of evolutionary dynamics on any complex population structure reveals stem cell niche architecture as a spatial suppressor of selection

Yang Ping Kuo[1,2], César Nombela-Arrieta [3] & Oana Carja [1] ✉

How the spatial arrangement of a population shapes its evolutionary dynamics has been of long-standing interest in population genetics. Most previous studies assume a small number of demes or symmetrical structures that, most often, act as well-mixed populations. Other studies use network theory to study more heterogeneous spatial structures, however they usually assume small, regular networks, or strong constraints on the strength of selection considered. Here we build network generation algorithms, conduct evolutionary simulations and derive general analytic approximations for probabilities of fixation in populations with complex spatial structure. We build a unifying evolutionary theory across network families and derive the relevant selective parameter, which is a combination of network statistics, predictive of evolutionary dynamics. We also illustrate how to link this theory with novel datasets of spatial organization and use recent imaging data to build the cellular spatial networks of the stem cell niches of the bone marrow. Across a wide variety of parameters, we find these networks to be strong suppressors of selection, delaying mutation accumulation in this tissue. We also find that decreases in stem cell population size also decrease the suppression strength of the tissue spatial structure.

Novel microfluidics and organoid technologies[1,2] allow us to start building biological scaffolds that control the spatial topology of a molecular or cellular population. In order to make full use of these innovations, we need a rigorous theory of how the structure of a population shapes its future evolutionary dynamics. This will allow us to design structures that either amplify the selective benefit and spread of beneficial mutations, or structures that suppress the spread of deleterious variants.

There is a large body of literature in population genetics theory studying the role of population structure in shaping evolutionary outcome, starting from the classic 1975 paper of Slatkin and Maruyama[3]. However, most previous modeling approaches that incorporate spatial patterns of variation usually only assume a few demes (patches) and symmetrical structures[4–6], simple topologies that can be embedded into two-dimensional continuous Euclidean space. In most cases, these simple topologies do not change fixation probabilities and rates of evolution compared to well-mixed populations[7,8]. These symmetrical structures fundamentally fail to capture the complex pattern of interaction and the variance in local selection pressure present in natural populations, as well as in emerging spatial cellular and molecular atlases[9–11].

[1]Computational Biology Department, School of Computer Science, Carnegie Mellon University, Pittsburgh, PA, USA. [2]Joint Carnegie Mellon University-University of Pittsburgh Ph.D. Program in Computational Biology, Carnegie Mellon University, Pittsburgh, PA, USA. [3]Department of Medical Oncology and Hematology, University and University Hospital Zurich, Zurich, Switzerland. ✉e-mail: oana@cmu.edu

Studying more complex topologies, ones for which there exists no homeomorphism to the well-behaved two-dimensional Euclidian space, becomes a much harder mathematical problem. These topologies can be represented using networks and we can use the mathematical formalism of the Moran birth-death process on graphs[12,13] to explore how spatially-structured patterns of interaction and replacement drive the composition of populations and shape the outcome of the evolutionary process. Under these models, a population of individuals is located on nodes of a graph and the links of a node indicate the neighboring nodes that can be replaced by its offspring[12]. Graph theory has been successfully used to study patterns of spatial variation and interaction across a wide range of scientific fields, from the social sciences to brain science[14–17].

Initial studies on very small graphs observe large differences between networks in the fixation probability of new mutants, compared to well-mixed populations[12,18], a marked departure from previous deme-based models. Some graphs are suppressors of selection, graphs that reduce the fixation probability of advantageous mutations, while increasing it for deleterious mutants[12,18,19]. Other graphs can be classified as amplifiers, increasing rates of evolution. One of the first general results, the isothermal theorem, states that in graphs where the propensity for change in each node is exactly the same, the fixation probabilities of new mutations are the same as in well-mixed populations. The assumptions of the isothermal theorem, however, sit on a knife edge; make small perturbations to the network structure and the assumptions no longer hold[12]. While these initial studies hint at the promise of graph theoretical approaches, analytic results with predictive power have been very difficult to derive. Most prior results either rely on very small networks (where build and solve time scale exponentially with population size, making them unsuitable for the study of networks of more than 30 nodes[19]) or invading mutants in the limit of neutrality, results that do not scale to generality[20]. What are the probabilities of fixation for a new mutation as a function of where and when it appears in much larger, and more biologically-realistic spatial networks?

Here, we develop an analytic approach that gives us the ability to systematically study probabilities of fixation on larger, heterogeneous spatial structures and identify graph properties that control suppression or amplification of selection, either leading evolution to a standstill or accelerating the evolutionary process. Linking network topology to evolutionary dynamics is complicated by the fact that networks differ in many structural properties. We build graph generation computational methods that allow the ability to systematically tune network statistics and study their role on probabilities and times to fixation for new mutations in the population, mathematical proxies for evolutionary outcome. Our algorithms combine simulated annealing procedures and degree preserving edge swapping[21] to continuously tune network properties one at a time, while keeping other properties constant. This allows us to fully understand the role of specific network parameters, as well as translate the meaning of the relevant parameters for any given network, across different graph-families.

Using our simulations and analytical approximations, we find that knowing the degree distribution alone is not enough to determine the fixation probabilities and times to fixation, since graphs with the same degree distribution, but different mixing pattern can exhibit very different evolutionary outcome. Importantly, we analytically derive the relevant selective parameter for a given network, without making restrictive assumptions on network type, size or selective advantage of the new invading variant.

In addition to the purely theoretical interest of the questions presented above, we also showcase how our theoretical results can be used to analyze rates of evolution in the stem cell populations of the bone marrow[22,23]. We use recent imaging data sets[24,25] to build the spatial stem networks of the bone marrow and we find that these networks are strong suppressors of selection, across a wide range of parameter choices and regardless of the type of the assumed birth-death process. Moreover, we find decreasing suppression with decreasing population size, hinting at a potential decrease in the suppressive properties of the spatial structure as individuals age.

## Results
### Model
We use a Moran-type model to describe changes in allele frequencies in a finite population of constant size $N$. Each individual's genotype is defined by a single biallelic locus $A/a$, which controls the individual's reproductive fitness. An individual with the $A$ allele is assumed to have fitness one, while an individual with allele $a$ has assigned fitness $(1 + s)$.

We use the structure of a graph to represent the structure of reproduction and replacement of the population. Every individual occupies a node in the graph, while the edges between nodes represent the local pattern of replacement. At every generation, we update the allele frequencies using two different update scenarios (Fig. 1A). In the first update scenario, we assume reproduction occurs before death (the Birth-death $Bd$ scenario). At every time step, we first select one individual for reproduction, with probability proportional to fitness, from the entire population. We then randomly select one of its neighbors for death and vacate the node for the new offspring. In the second update scenario, denoted as the death-Birth $dB$ update, we first select a node at random from the population to be vacated and then choose one of its neighbors for reproduction, with probability proportional to fitness. Note that selection happens only when choosing the individual to reproduce. This means that, in the Birth-death update, the individuals compete globally, at the population level, while in the death-Birth update, the selection step is local, the competing individuals are only the neighbors of the node randomly chosen for death. Due to these differences in global versus local competition, the two update rules have been shown to lead to drastically different evolutionary dynamics[12,18].

The graph structure therefore becomes a mathematical proxy for the spatial topology or the population structure of replacement: individuals reproduce locally, and their offspring spread to neighboring nodes connected by an edge. The graphs we consider here are unweighted and undirected. Initially, we assume the population fixed on the wild-type $A$ allele. We introduce one mutant $a$ individual at a random node at time $t = 0$ and we ignore subsequent mutation. Under this model, the population will eventually reach a monomorphic state where individuals of the same $A/a$ allele occupy all nodes in the graph. We study the probability of fixation of the invading allele $a$ as a function of the population size $N$, the selective coefficient of the new mutant $s$ and importantly, the topological features of the network spatial structure. Our goal is to systematically study the role of the network structure in shaping rates of evolution by directly comparing these probabilities of fixation with the equivalent probabilities in a well-mixed population.

Linking network topology to evolutionary dynamics is complicated by the fact that networks differ in many structural properties and tuning parameters independent of others is not a trivial problem. To identify the relevant graph properties that either speed up or suppress adaptation through shaping probabilities and times to fixation of new mutants in the population, we characterize graphs through the lens of their main two components: the nodes and the edges. We can therefore think of graph properties as either node- or edge-centric. The main property of a node is the node degree (the number of neighbors the node is linked to) and the node degree distribution becomes an important global network property[26]. Graph edges, on the other hand, can be categorized based on the type of nodes they connect and how often they connect nodes of different degrees. The mixing pattern of a graph (also called graph assortativity) is a global edge-centric graph descriptor that informs on the frequencies of each edge type[27].

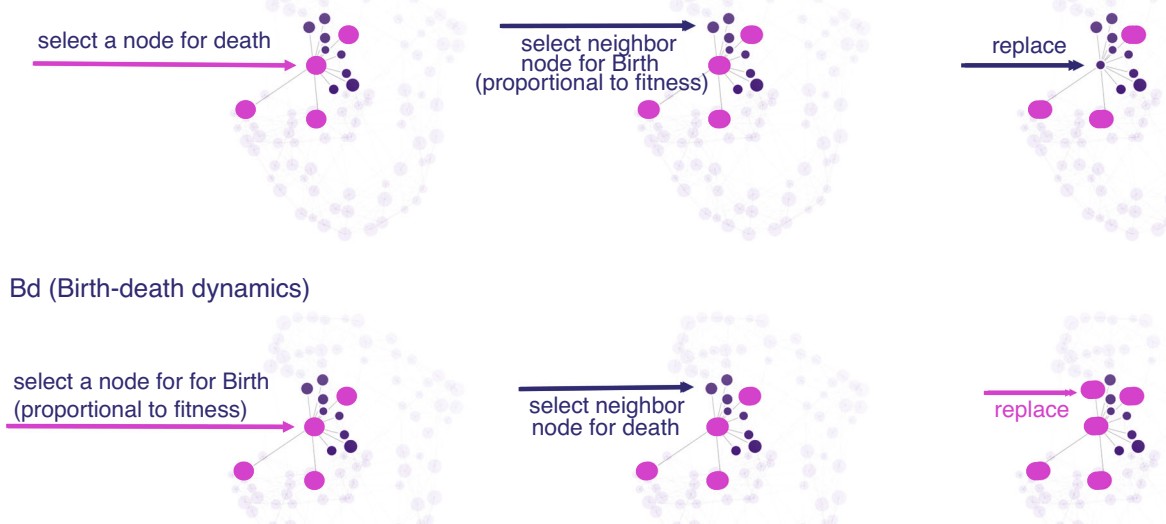

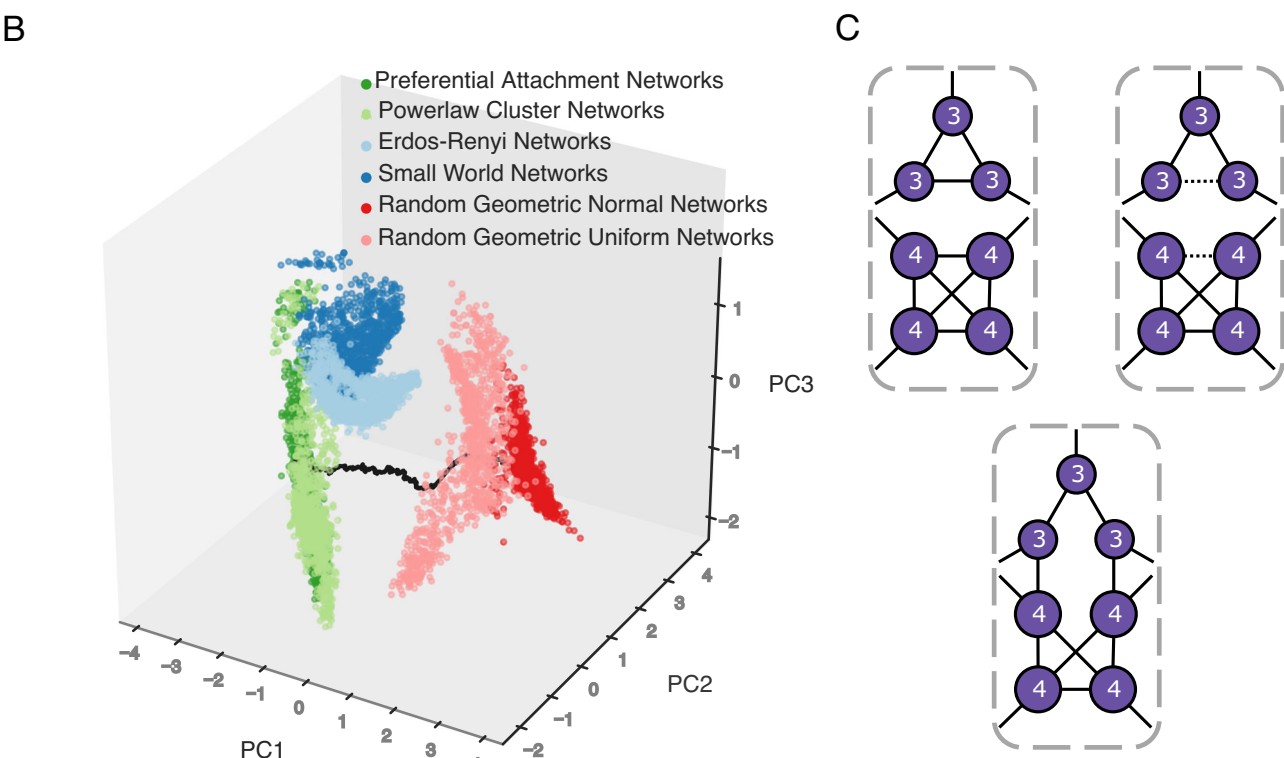

**Fig. 1 | Illustration of the population birth death process update rules and graph rewiring methods. A** illustrates both the dB (death-Birth) and the Bd (Birth-death) update rules. In (**B**), we use principle component analysis on 6 graph characteristics (mean, variance, third moment, modularity, average clustering, and assortativity) to highlight the network families studied, as well as the trajectories between them. Each graph family shows clustering using the first three principle components (that explain 89% of the variance in PC space). The black line represents the trajectory in PC space as we rewire graphs starting from preferential attachment (PA) graphs, through power law cluster networks (PLC) and uniform random geometric graphs, to normal random geometric graphs (RGG). **C** illustrates the edge swap operation used to tune graph characteristics. At first, there are no edges connecting nodes of degree 3 to nodes of degree 4. Two edges are randomly selected to be disconnected and nodes that were "parallel" with respect to the two disconnected edges are then connected, thus preserving the number of edges. After the rewiring step, there are two edges connecting nodes of degree 3 to nodes of degree 4, and there is no longer a 4-clique. The degrees of the nodes, however, are preserved.

To begin with, we use random graph generators and construct graphs that span the known graph families[28–34]. This ensures that our results are generalizable across graph families and graph properties (Fig. 1B). Existing graph generating methods, however, do not allow for the separate tuning of the degree distribution and the node degree mixing pattern. For example, in preferential attachment (PA) graphs, one can smoothly change the shape of the degree distribution by changing the power of preferential attachment $\beta$. However, if we change this preference of connection to high degree nodes, we inevitably also change the graph's mixing pattern and cannot independently study its role in amplifying or suppressing selection.

To allow for tuning of mixing patterns, independent of the degree distribution, we implement a sampling network generation algorithm based on simulated annealing (Fig. 1C). See the "Methods" section for more information on the algorithm and the graph families generated and used in this study.

Once the network structure is set, we use ensembles of at least 10,000 Monte Carlo simulations, as well as analytic approaches as described in the next section, to compute the probabilities of fixation of the new allele $a$. We study the probability of fixation of a new invader mutation $a$ with fitness $(1 + s)$ that appears in a random initial node of the network and compare it with the equivalent probabilities of fixation in well-mixed populations. We start by obtaining analytic approximations for the dB (death-Birth) update model and then discuss the important differences specific for the Bd (Birth-death) update rule. Intuitively, the dB dynamics applies when the evolutionary update are driven by available space being freed up, followed by local competition among the neighbors, whereas the Bd dynamics applies in cases where competition happens globally, but replacement is driven by the local pattern of interaction.

## Analytic description

We begin by presenting the main ideas of our analytic approximation for the probability of fixation of the $a$ allele. For the complete mathematical treatment, please see Supplementary Notes 1 and 2. Previous analytic approaches have either made use of the adjacency matrix of the network (which uniquely identifies the graph) and its associated transition probabilities[19] or assumed that the evolutionary dynamics are in the limit of neutrality and a vanishing selection coefficient $s$ (if weak selection is assumed, the probability of fixation can be approximated by treating it as the linear perturbation to the continuous coalescence, the dual of the Moran process under neutrality)[20,35]. The former approach can provide closed form solutions for the fixation probability of $a$, but becomes intractable for large networks since it tracks a Moran process with $2^N$ states and the algorithm build and solve time both grow exponentially with population size (even for $N = 23$ nodes, the approach becomes unfeasible[19]). The latter approach reduces the problem from exponential to polynomial complexity in population size $N$[20,35], however it performs poorly as we move away from the neutrality limit for the $a$ allele (Supplementary Figs. S1 and S2).

The approach we take here is to use the node degree distribution, and only keep track of the mutant frequencies $x_i$ at all $N_i$ nodes of the same degree $d_i$. Let $D = \{d_1, d_2, …, d_i, …\}$ represent the set of all possible node degrees. While the degree distribution might not uniquely represent the network and some of the graph information is lost, this approach nonetheless greatly reduces the number of possible states in the Moran model[17,36]. We denote the frequency of nodes of degree $d_i$ in the population by $p_i$. To model node degree mixing, we use $p_{ij}$ to denote the probability that a node of degree $d_i$ is connected to a node of degree $d_j$. The probability of fixation of allele $a$ is then approximated using the diffusion approximation[37,38].

At every time point, $x_i$, the frequency of the mutant at nodes of degree $d_i$, increases by $1/N_i$ with probability $T_i^+$ and decreases by $1/N_i$

with probability $T_i^-$. We can write:

$$
\begin{aligned}
T_i^+ &= \frac{1+s}{W} \sum_{j \in D} p_j p_{ji} x_j (1 - x_i) \\
T_i^- &= \frac{1}{W} \sum_{j \in D} p_j p_{ji} (1 - x_j) x_i,
\end{aligned}
\tag{1}
$$

where $W$ is the mean fitness of the individuals in the population.

We use these transition probabilities to find the mean and covariance of the change in $x_i$ per unit time and use the backward Kolmogorov equation[38] to find the probability of fixation of the $a$ allele for any initial mutant frequency $P(\vec{x})$:

$$
\sum_i \left( \frac{T_i^+ - T_i^-}{p_i} \frac{\partial P}{\partial x_i} + \frac{1}{2} \frac{T_i^+ + T_i^-}{N p_i^2} \frac{\partial^2 P}{\partial x_i^2} \right) \\
- \frac{1}{2} \sum_{i,k} \frac{(T_i^+ - T_i^-)(T_k^+ - T_k^-)}{N p_i p_k} \frac{\partial^2 P}{\partial x_i \partial x_k} = 0.
\tag{2}
$$

Here, the coefficient for the linear differential operator is quadratic in $x_i$ and the coefficient for the quadratic differential operator is quartic in $x_i$.

By using singular perturbation to linearize the coefficients of the differential equation[39], the solution to the partial differential equation in (2) for the Birth-death update model can be approximated using:

$$
\sum_{i,j \in D} p_i p_{ij} \left( \frac{1}{2 p_i^2} ((1+s)x_j + x_i) \frac{\partial^2 P}{\partial x_i^2} + \frac{1}{p_i} ((1+s)x_j - x_i) \frac{\partial P}{\partial x_i} \right) = 0.
\tag{3}
$$

The death-Birth process shares a similar equation, given by:

$$
\sum_{i,j \in D} p_j p_{ji} \left( \frac{1}{2 p_i^2} ((1+s)x_j + x_i) \frac{\partial^2 P}{\partial x_i^2} + \frac{1}{p_i} ((1+s)x_j - x_i) \frac{\partial P}{\partial x_i} \right) = 0.
\tag{4}
$$

The only difference between the two equations is the change from $p_i p_{ij}$ to $p_j p_{ji}$. The solution for the Bd process can then be written as:

$$
P(\vec{x}) = \frac{1 - \exp\{-N \sum_{i \in D} p_i A_{Bd,i} x_i\}}{1 - \exp\{-N \sum_{i \in D} p_i A_{Bd,i}\}}.
\tag{5}
$$

We can compute $A_{Bd}$ by solving the following system of quadratic equations:

$$
\sum_{j \in D} \left[ (1+s) A_{Bd,j}^2 p_i p_{ij} + A_{Bd,i}^2 p_j p_{ji} - 2(1+s) A_{Bd,j} p_i p_{ij} + 2 A_{Bd,i} p_j p_{ji} \right] = 0, \quad \forall i
\tag{6}
$$

while for dB update processes we need to solve:

$$
\sum_{j \in D} \left[ (1+s) A_{dB,j}^2 p_j p_{ji} + A_{dB,i}^2 p_i p_{ij} - 2(1+s) A_{dB,j} p_j p_{ji} + 2 A_{dB,i} p_i p_{ij} \right] = 0, \quad \forall i.
\tag{7}
$$

In the death-Birth process, the contribution to the fixation probability due to the degree distribution is on the order of the selection coefficient $s$, while the contribution due to degree mixing is on the order of $s^2$. Therefore, knowing the degree distribution of the graph gives a good approximation to the probability of fixation, for weak $s$. Assuming $s \sim \frac{1}{N}$, the probability of fixation can be approximated as:

$$
P_{dB} = \frac{1 - e^{-\alpha_{dB} s/(1+s/2)}}{1 - e^{-\alpha_{dB} N s/(1+s/2)}}, \quad \text{where } \alpha_{dB} = \frac{\langle d \rangle^2}{\langle d^2 \rangle}.
\tag{8}
$$

Here, $\langle d \rangle = \sum p_i d_i$ and $\langle d^2 \rangle = \sum p_i d_i^2$ are the first and second moment of the degree distribution. This selection suppression or

amplification factor $\alpha_{dB}$ can be used to measure how much the probability of fixation differs from that of well-mixed populations. If $\alpha = 1$, the probability of fixation is identical to well-mixed populations. Graphs with $\alpha > 1$ are amplifiers and $\alpha < 1$ are suppressors. In the limit of weak selection, Eq. (8) becomes $\frac{1-e^{-s}}{1-e^{-Ns}}$ for the well-mixed population[40,41].

Our approximation shows that, for sufficiently weak selection, $\alpha_{dB}$, the suppression parameter for dB processes, is a function of the first and second moments of the degree distribution alone.

Solving Eqs. (6) and (7), we show the accuracy of the analytic approximation in Fig. 2 using preferential attachment PA graphs. As the mean of the degree distribution increases, probability and time of fixation increase for the death-Birth process (and decrease for the Birth-death process) toward the well-mixed population limit. This makes intuitive sense: as the mean degree increases, the graph structure approaches that of a well-mixed population. In contrast, as the variance of the degree distribution increases, while keeping the mean constant, probabilities and times to fixation decrease monotonically

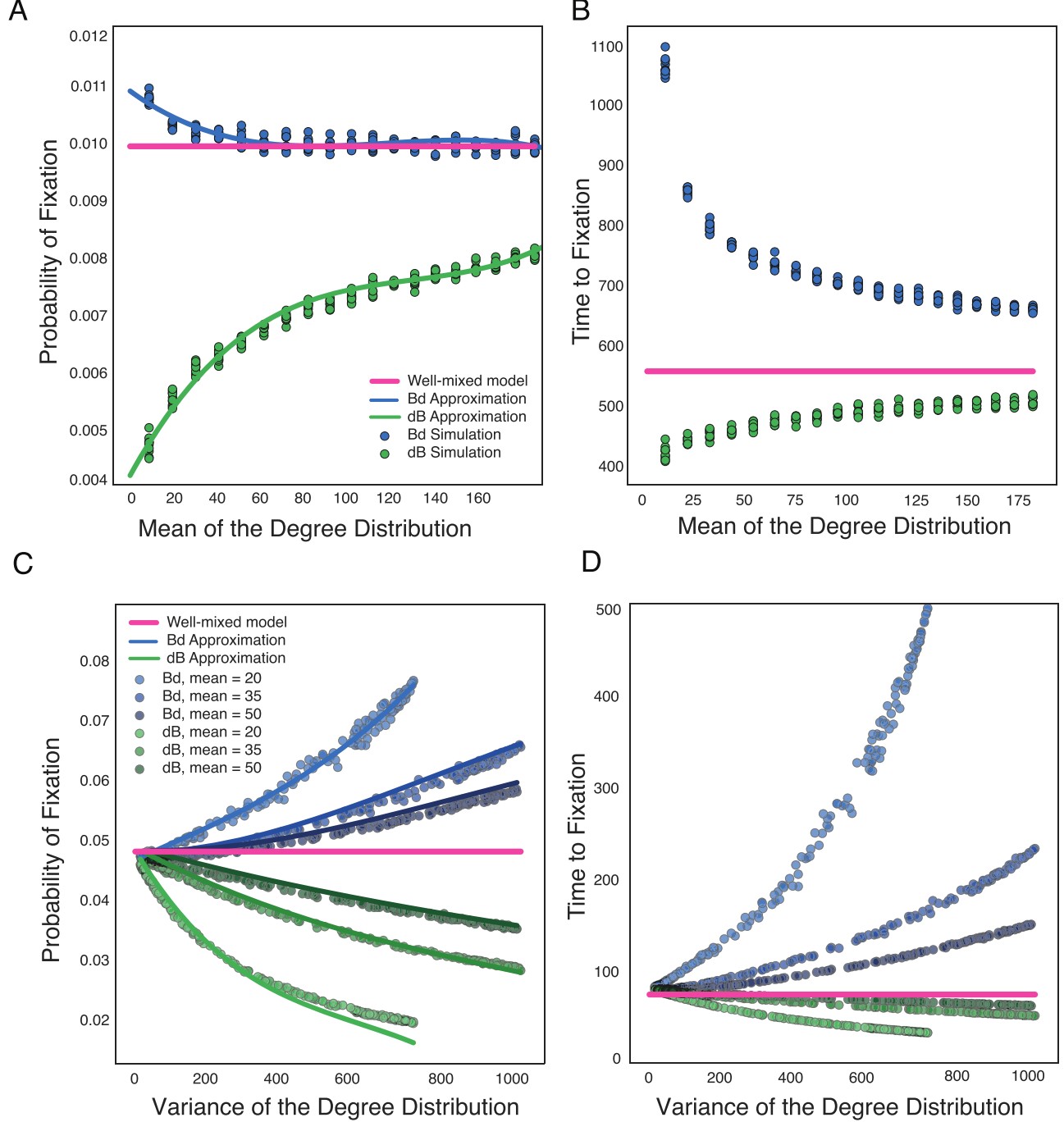

**Fig. 2 | Role of the first moments of the degree distribution on evolutionary dynamics.** The dots represent ensemble averages across $10^6$ replicate Monte Carlo simulations, while the lines represent our analytical approximations. **A, B** We plot the probabilities and times to fixation as a function of the mean of the degree distribution. We use preferential attachment PA graphs, graph size $N = 1000$ and $Ns = 10$. **C, D** We plot the probabilities and times to fixation as a function of the variance of the degree distribution, while keeping mean degree constant, as in the legend. Here, graphs are PA graphs, $N = 100$ and $Ns = 5$. Probability and time to fixation for well-mixed model are calculated using methods outlined in ref. 40 (see the "Analytic description" subsection).

for the dB process, and increase for the Bd process. The variance measures how heterogeneous the nodes are. At variance zero, all the nodes in the graph have the same number of neighbors, which means the graph is isothermal and the fixation probability is the same as that of well-mixed populations.

Using the approximation in (8) for the death-Birth dB process, we show the probability of fixation of a new mutation $a$ across multiple graph families in Fig. 3A. As the effective selection parameter $\alpha$ increases, the fixation probability increases, reaching and crossing the well-mixed line when $\alpha$ equals to one. Intuitively, $\alpha$ quantifies the interplay between the mean and variance of the degree distribution, between how well-connected the nodes are and the network node heterogeneity.

### The evolutionary role of graph mixing pattern

For the Birth-death process, unlike the case of the death-Birth process where the effects of mixing pattern can be ignored under weak selection, network degree distribution and mixing pattern both contribute to the new mutation's fixation probability. Similar to the death-Birth process, the contribution to the fixation probability due to degree distribution is again on the order of the selection coefficient of the new mutation $s$. However, in contrast to the death-Birth process, the graph mixing pattern has the same order of magnitude contribution as the graph degree distribution. Under selection $s \sim \frac{1}{N}$, the probability of fixation can be approximated as:

$$P_{Bd} = \frac{1 - e^{-\alpha_{Bd}s/(1+s/2)}}{1 - e^{-\alpha_{Bd}Ns/(1+s/2)}}, \quad \text{where } \alpha_{Bd} = \left(\langle d^{-1}\rangle \sum_{i,j\in D} p_j p_{ji} d_i^{-1}\right)\left(\sum_{i,j\in D} p_j p_{ji} d_i^{-2}\right)^{-1}.$$

$$(9)$$

Here, $\langle d^{-1}\rangle = \sum p_i d_i^{-1}$ is the first inverse moment of the degree distribution.

For the Birth-death process, the $\alpha_{Bd}$ selection factor can be written as a function of parameters of the network wiring pattern and properties of its degree distribution. This approximation is shown in Fig. 3B, alongside the results of Monte Carlo simulations. The fixation probability increases as the selection parameter $\alpha$ increases, with lower values for random geometric graphs and higher selection amplification for the preferential attachment graph family.

To understand the underlying network properties controlling evolutionary dynamics of new mutations, we need an intuitive understanding of the amplification factor in Eq. (9). The inverse moment $\langle d^{-1}\rangle$ quantifies the shape of the degree distribution, while the rest of the parameters in Eq. (9) can be thought of as parameters that measure the graph's assortativity or mixing pattern (ref. 27). A graph is assortative when a node of degree $d_i$ preferentially attaches to other nodes of a degree similar to $d_i$. A graph is called disassortative when the number of edges that connects nodes of degree $d_i$ and nodes of dissimilar degree is higher than the expected number in randomly mixing graphs. Consider an edge swapping operation on a graph that breaks two edges: one between two nodes of degree $d_i$ and one between two nodes of degree $d_j$. Two edges that connect node of degree $d_i$ and degree $d_j$ are formed from the stubs. If $d_i$ and $d_j$ are dissimilar, such a rewiring step reduces the graph's assortativity. Assuming the population size is large, the change in the $\alpha_{Bd}$ amplification factor can be written as:

$$\Delta\alpha_{Bd} \sim \left(\frac{1}{d_i} - \frac{1}{d_j}\right)^2 \left(\frac{1}{d_i} + \frac{1}{d_j} - \frac{\mu_2}{\mu_1}\right), \text{where } \mu_1 = \sum_{i,j\in D} p_j p_{ji} d_i^{-1} \text{ and } \mu_2 = \sum_{i,j\in D} p_j p_{ji} d_i^{-2}.$$

$$(10)$$

More details on this derivation are given in Supplementary Note 3. The magnitude of the change depends on the difference between the reciprocals of the degrees. If new edges are created between nodes of very dissimilar degrees, the change in the fixation probability can be significant. Since the change depends on the reciprocal, nodes of low degree have a disproportional effect on the change in amplification. The upper bound of $\mu_2/\mu_1$ is $1/d_{\min}$, where $d_{\min}$ is the smallest degree of the graph. This means that if either $d_i$ or $d_j$ is close to the lowest degree, $\alpha_{Bd}$ is guaranteed to increase. In other words, the probability of fixation increases when there are more edges connecting nodes of low degrees to nodes of high degrees (disassortative graphs). An example of this is the star network, one of the strongest known amplifiers for undirected graphs[42–44]. A star graph consists of a few nodes forming the center, while the rest of the nodes connect to the center nodes and form the vertices of the star. As a consequence, the nodes at the center have high degrees, while the rest tend to have significantly smaller degree, and the only edge type in the graph is between nodes of very different

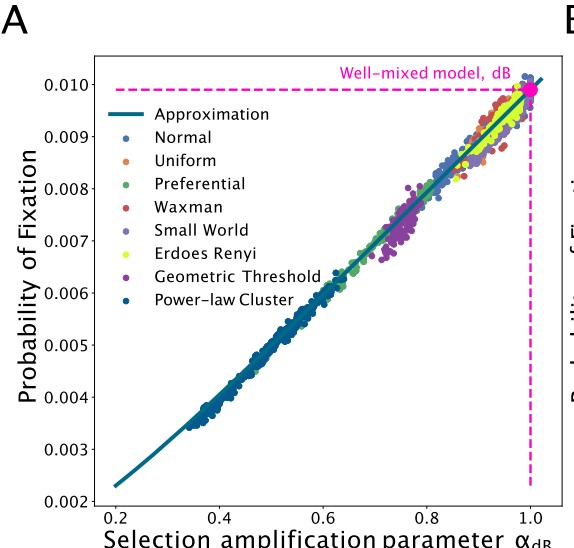

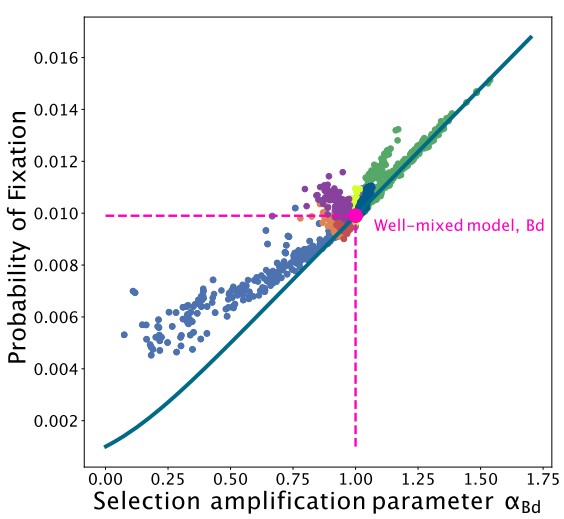

**Fig. 3 | Probabilities of fixation across graph families.** The fixation probability is shown on the *y*-axis as we vary the evolutionary quantity $\alpha$ on the *x*-axis. The dots represent ensemble averages across $10^6$ replicate Monte Carlo simulations, while the lines represent our analytical approximations. Here, graph size $N = 1000$ and

$Ns = 10$. Each dot represents simulations on a distinct graph. The various colors represent graphs generated using different generation algorithms. **A** shows results for the death-Birth process. **B** shows results for the Birth-death process.

degrees. This type of graphs have the highest disassortativity, and strongest amplification of selection.

We can use this intuition to also explain the relative location of graph families in Fig. 3B. Preferential attachment networks tend to be graphs with low assortativity (high disassortativity), with many hub-and-spoke structures (lower degree nodes connected to high degree nodes) and thus strong amplifiers. In contrast, normal geometric graphs with non-uniform spatial density tend to have high assortativity and thus tend to suppress the force of selection. This is due to the fact that nodes in high-density areas tend to be closer to each other and, assuming the density function is relatively smooth, neighbors tend to have similar degrees. Similarly, nodes in low spatial density areas tend to have few neighbors (low degrees), and so do their neighbors. Therefore, in spatial graphs with nonuniform spatial density, nodes are connected with neighbors of similar degree. This explains the suppression effect of this network family.

Since knowing the degree distribution alone is not enough to determine the fixation probability and amplification parameter for the Birth-death process, to illustrate the effects of assortativity and mixing pattern on fixation probabilities without the influence of degree distribution and graph generating method, we use edge swap operations to sample graphs with different mixing patterns, while keeping the degree distribution the same. We use graphs generated from different generating methods as input graphs to ensure generalization across graph families. For the same degree distribution, the spread of values for the fixation probability due to the effect of degree mixing can be substantial (Fig. 4A). Here, all dots of the same color represent graphs from the same starting graph family and are altered by the edge swap sampling method with different end mixing patterns. We use the variance of the degree distribution as a measure for the shape of the distribution. Although the mean degree is not shown, dots of the same color share the same mean.

We use degree Pearson correlation $r$ as a measure of the mixing pattern in the graph. We maximize and minimize the degree correlation to obtain an ensemble of graphs with the same degree distribution

but different mixing pattern. When $r = 1$, the network has perfect assortative mixing patterns, while $r = -1$ corresponds to the case of a disassortative network. The contribution of the mixing pattern in the amplification constant in Eq. (9) is also a measure of assortativity (Fig. 4B). $r$ and $\alpha_{Bd}$ have a negative correlation, as expected. For graphs with the same degree distribution, the graphs that have low assortativity (high $\alpha_{Bd}$) have a higher probability of fixation (Fig. 4C). The difference between Figs. 4C and 3B is that in 4C we keep the degree distribution constant. Therefore, both node types and edge types in a graph both contribute to evolutionary dynamics on the graph structure.

## Increased suppression of selection in large populations

While it has been previously claimed that under the Birth-death process most graphs are amplifiers of selection[18], our results above show that a large fraction of Birth-death graphs are suppressors of selection. The discrepancy in the results is due to the different population sizes considered. Due to computational and analytic limitations, previous studies consider very small population sizes of under $N = 30$ individuals. In this section, we study the effects of population size on fixation probabilities using two types of graphs: star graphs, known to be one of the strongest undirected amplifiers, and detour graphs, strong suppresors[43,44].

A detour graph consists of a completely connected central cluster and a cycle part (see Fig. 5). These graphs have a low probability of fixation due to their high assortativity, since the graphs only have two edges connecting nodes of different degrees. We show that the fixation probability depends on the size of the central cluster, i.e., the length of the detour. To find the cluster size that minimizes the probability of fixation, we use the solution to the diffusion Eq. (47) from Supplementary Note 2, derived using regular perturbation (ref. 45):

$$P_{Bd} \approx \frac{1}{N} + s\sum_{ij}p_i p_j A_{ij}, \qquad (11)$$

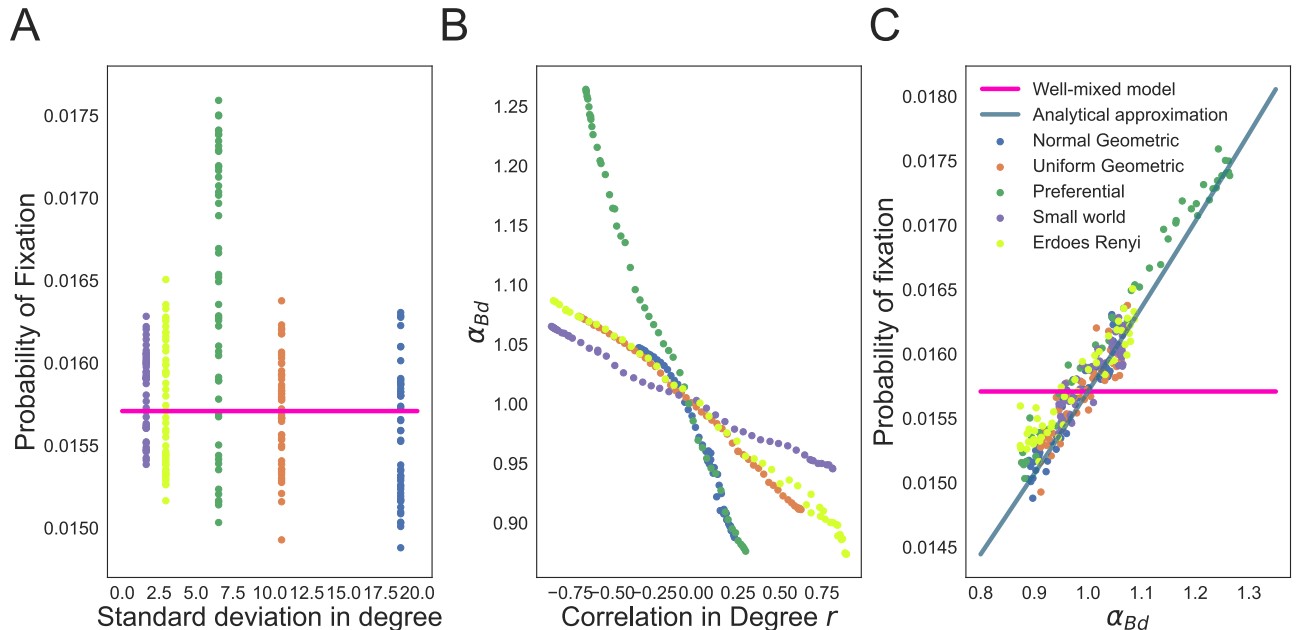

**Fig. 4 | Effects of the mixing pattern on the Bd probability of fixation.** The dots represent ensemble averages across $5 \times 10^6$ replicate Monte Carlo simulations, while the lines represent our analytical approximations. Here the degree distribution is held constant as we vary the mixing pattern of the graphs, $N = 100$ and $s = 0.1$. Colors indicate the graph family (with the same degree distribution) as in the legend. Mixing pattern of the graph is tuned using edge swapping operations. To highlight that mean and variance in degree is not enough to predict probabilities of fixation, **A** shows that the probability of fixation can span a wide range of values, for the same mean and standard deviation in degree. **B** shows the dependence of the amplification parameter on the mixing pattern, or correlation in degree $r$. **C** shows that the selection amplification factor from Eq. (9) explains the effect of graph mixing pattern on the probability of fixation.

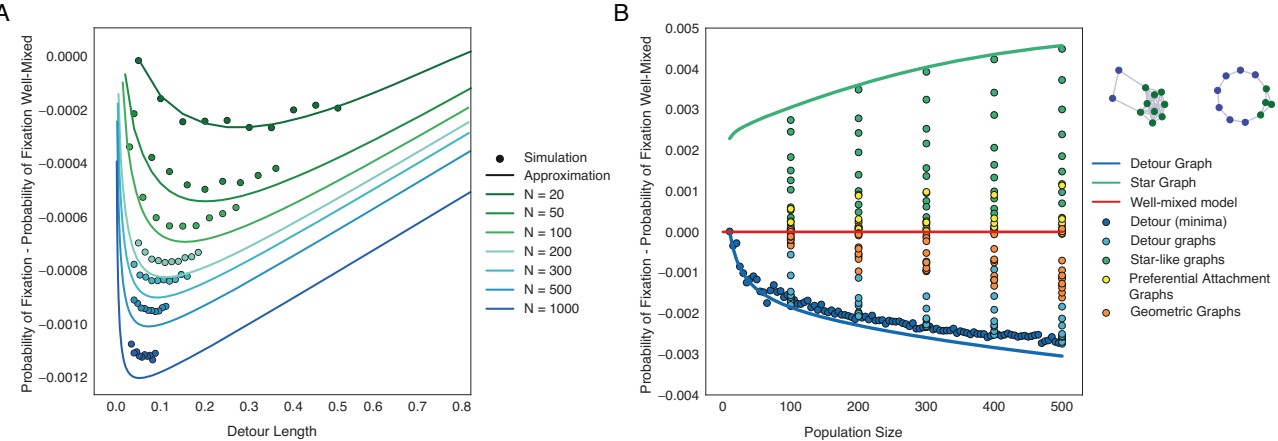

**Fig. 5 | Detour and star graphs are graph families that minimize and maximize of probability of fixation under the Bd process.** The dots represent ensemble averages across $10^6$ replicate Monte Carlo simulations, while the lines represent our analytical approximations. Detour graphs are the strongest suppressors under weak selection. Detour graph consists of a completely connected central cluster and a cycle part. In (**A**), we plot the probability of fixation on detour graphs with varying lengths of the detour cycle. Here $s = 0.002$. In (**B**), we plot the probability of fixation across graph types, with respect to population size, to illustrate how the star and detour graphs represent extreme values of the fixation probability. Here $s = 0.005$. The two graphs illustrated are detour graphs with different detour lengths.

where $A_{ij}$ satisfy the following system of linear equations (see more detail on the derivation in Supplementary Note 4):

$$p_j p_{ji} \left( -\frac{1}{\langle d^{-1} \rangle d_i} + 2\frac{A_{ii}}{N} \right) - 2\sum_k p_i p_j p_{jk} A_{ki} + 2\sum_k p_j p_k p_{ki} A_{ij}$$
$$+ p_i p_{ij} \left( -\frac{1}{\langle d^{-1} \rangle d_j} + 2\frac{A_{jj}}{N} \right) - 2\sum_k p_j p_i p_{ik} A_{kj} + 2\sum_k p_i p_k p_{kj} A_{ij} = 0. \tag{12}$$

The size of this system of equations is $|D|(|D| + 1)/2$, where $|D|$ is the number of unique degrees in the graph. Since a detour graph has only two types of degrees, we only need to solve a system of 3 equations for $A_{11}$, $A_{12}$, and $A_{22}$. The only variable that influences the probability of fixation in the detour graphs is the length of the detour. We plot the difference in probabilities of fixation for detour graphs of different sizes and the well-mixed population against the length of the detour in Fig. 5A. When detour length equals zero, we have the complete graph where the difference in the probability of fixation is zero. Since detour length is maximized in a ring graph, the probability of fixation initially decreases as the length of the detour is increased, reaching a minimum, before increasing toward the well-mixed control. It can also be observed that the minimum decreases with population size. The regular perturbation approximation is used instead of Eq. (9) since, while the approximation predicts the magnitude of suppression on detour graphs, the minima are shifted slightly to the left toward well-mixed. Mathematically, this is due to the fact that singular perturbation deviates from the solution of the diffusion equation when the exchange of individuals between two sub-populations is weak[46].

The star graphs and the detour graphs constitute limiting structures for the range of probabilities of fixation for undirected graphs under sufficiently weak selection (Fig. 5B). The difference in probability of fixation between the detour graph and a well-mixed population is close to zero when graph size is small, however it decreases sharply as population size increases. This explains why strong suppressors are prevalent in large populations, but rarely observed in small populations. Although we did not rigorously prove that detour graphs serve as the lower bound for the probability of fixation under the Birth-death update, this is empirically observed in graphs of small size (ref. 43). It is reasonable to assume the existence of large graphs that have stronger suppression, but this only reinforces our point that

suppressors are more prevalent in larger populations. The result is particularly biologically interesting. Imagine populations with individuals fixed in space, such as species in an ecosystem or cells in biological tissues. These spatial populations can be reasonably approximated by random geometric or Waxman graphs. As shown in the previous section, these types of populations are likely to be suppressors under the Birth-death update. If the size of the population were to decrease (for example, environmental catastrophes or injury and aging of tissues), not only will the force of drift increase in the population, but also the suppressive capability of the population against the invasion of beneficial mutation will be compromised. This could lead to increased likelihood of beneficial mutations propagating in the population (and potential rescue the population from extinction), or increased rate of accumulation of deleterious driver mutations that initiate neoplasms.

## Application to mutation accumulation in hematopoietic stem cell populations

We show how the theory developed above can be linked to novel datasets of spatial localization and specifically, be used for the study of rates of mutation accumulation in the hematopoietic stem cell (HSC) population of the bone marrow. Hematopoietic stem cells reside in specialized micro-environments, or niches, where distinct mesenchymal cells, the vasculature, and differentiated hematopoietic cells interact to regulate stem cell maintenance and differentiation (refs. 47,48). These niches are fixed in location and number, with heterogeneous spatial structure, and stem cells are in constant competition for niche occupancy (refs. 49,50).

New innovations in imaging techniques and our ability to process these images at scale offer unprecedented opportunities to study how the spatial heterogeneity of stem cell niches shape tissue evolutionary dynamics. Just as demographic surveys can reveal the rates at which a contagious disease can spread through a spatially heterogenous population, these imaging datasets allow us to quantify cellular and molecular patterns of spatial variation and study how these topologies shape evolutionary dynamics. We use published datasets that provide the spatial location of hematopoietic stem and progenitor cells in four samples of mouse tibia[24] and the spatial locations of 8 bone marrow samples of CXCL12-abundant reticular cells (which critically modulate hematopoiesis at various levels, including hematopoietic stem cell maintenance), each with two images of two anatomically distinct

regions (the diaphysis and the metaphysis), in total 16 cellular populations[25].

Adult hematopoietic stem cells are known to divide symmetrically, whereby a mother stem cell either divides into two differentiated daughter cells or two undifferentiated stem cells[51,52]. The two modes of symmetric division are analogous to birth and death in a population described by the Moran process. We build the networks of stem cell niche architecture and use the inferred spatial topologies to infer the accumulation rate of driver mutations, the main cause of cancer in cycling tissues[22] (see "Methods" section). One resulting network is shown in Fig. 6A. For illustration purposes, for the network shown, the cutoff distance is set to 300 μm (4.78 × the distance between shortest pairs).

The difference in probabilities of fixation, compared to those in well-mixed populations, is plotted against the population size in Fig. 6B (Birth-death update) and 6C (death-Birth update). The color dots are the results using a cut-off distance of 15, which is the closest to our estimated biological interaction range (see the "Methods" section). We also show results for networks generated using cut-off radii ranging from 2 to 20 times the average distance between shortest pairs, as comparison. The fixation probabilities are either close to or lower than that of the well-mixed, except networks generated using two times the distance between the shortest pair as the cut-off radius (cut-off distance equal to 2).

We show that geometric graphs constructed from a non-uniform spatial distribution of individuals are likely to result in an assortative mixing pattern, hence we find suppression of selection in the hematopoietic stem cell populations, invariant to the underlying update process (compare panels 6B, C). We also construct networks with a probabilistic connection function[31] and observe no qualitative difference.

Our results also show that the strength of suppression increases as the stem cell population size increases and the fixation probability shows a negative correlation with population size (Pearson correlation of −0.687 and p value of 0.001). A similar conclusion is reached with most other cut-off distances (see examples of cut-off distances 10 and 20 in Supplementary Fig. S6), as well as other mutant selection coefficients (see examples of 5% and 10% fitness increases in Supplementary Fig. S7). A previous study by Dingli and Pacheco[53] predicts that the

total number of active stem cells in mammals scales with body mass with exponent 3/4. Assuming similar bone marrow tissue architecture in systems with more stem cells, suppression of selection is predicted to be amplified in larger mammals. This observation could partially explain the observed reduction in cancer incidence in large organisms as stated by Peto's paradox[54]. This also implies that processes such as injury or aging, that lead to reduced stem cell and niche count, could lead to the increased likelihood of beneficial mutations propagating in the population and an increased risk of developing cumulative diseases of aging, such as cancer.

## Discussion

Graphs represent a powerful tool to mathematically represent a population's structure of spread or interaction and to ask how properties of this structure shape the balance of evolutionary forces. However, obtaining closed form solutions for evolutionary dynamics on graphs has been particularly difficult. Here we introduce new theoretical and computational methods to rigorously study the role of graph topology on shaping evolutionary dynamics. We focus on parameters of the degree distribution and the graph mixing pattern because these distributions inform on graph-wide properties of the fundamental building blocks of a network: the nodes and the edges.

We show that the probability of fixation of a new mutation appearing on a random node can be approximated by solving a system of quadratic equations with number of variables depending on the number of degrees in the graph, which in most practical cases is efficient, even in large populations. By tuning the first moments of the degree distribution independently of each other, we analyze how the mean and variance in degree change probabilities and times to fixation. For example, we show that the probability of fixation under the Birth-death update increases monotonically as a function of the variance of the degree distribution. This is because the parameter that controls degree heterogeneity also controls the mixing pattern of the graph, by changing the connection bias toward nodes of higher degree.

Moreover, we write the relevant selective parameter of suppression or amplification ($\alpha_{dB}$ and $\alpha_{Bd}$), predictive of whether the network is an amplifier or suppressor of selection. While for the death-Birth process, this constant depends on properties of the degree distribution, for the Birth-death process, this constant is composed of

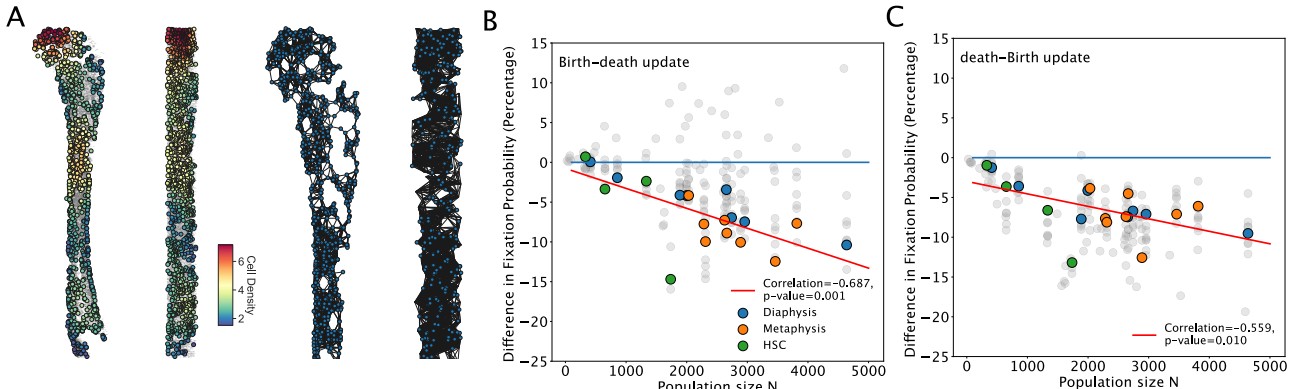

**Fig. 6 | The spatial networks of the hematopoietic stem cell (HSC) populations.** We use a public dataset of spatial locations in the bone marrow consisting of HSC locations (source data[24]) and proxy niche component locations (source data[25]). In (**A**), color dots represent an example of the spatial locations of stem cells in the cell population, while the colors corresponds to the density of cells. We show an example of the spatial distribution of hematopoietic stem cells in the mouse tibia: the *xy* cross-section of the tibia with cells counted in a 260 μm × 350 μm area and the *yz* cross-section of the tibia with cells counted in a 350 μm × 11.5 μm area. Using these data, we build the resulting geometric random graphs (the cut-off radius of the illustration is set to 300 μm for illustration purposes). In (**B, C**), we use these

networks to study evolutionary dynamics on these stem cell niche spatial networks as a function of their population size. The color dots use a cut-off distance of 15. Gray dots are results from other cut-off distances, ranging from 2 to 20, for comparison. The diaphysis is the shaft or central part of the tibia and the metaphysis is the neck portion of the bone (source data[25]). Here, *s* = 0.01 and *Ns* varies with population size. **B** shows results using the Birth-death process and **C** shows the death-Birth process. Dots are averaged over at least 1 million simulations. The *p* values are obtained from a Wald test with a null hypothesis that the slope is equal to zero.

parameters of both the mixing pattern and the degree distribution of the graph. If degree distribution is held constant, increasing the amplification parameter corresponds to increasing the disassortativity of the graph. Or reversely, increasing the disassortativity of a network increases the probabilities of fixation monotonically across multiple random graph families. For the death-Birth process, increasing disassortativity also increases the fixation probability, but only when selection is larger than order of $1/N$.

The limitation of this approach is that it ignores higher-order organizations of the network, such as community structures and network motifs and assumes that nodes of the same degree are topologically similar. Furthermore, our approach averages the assortativity of individual nodes over the entire graph and an example where this approach would not work is for a graph consisting of two parts connected by a few edges, one highly assortative, and one highly disassortative. The graph would be treated as neutrally assortative and this would lead to incorrect prediction of the fixation probability.

We also show that, contrary to prior empirical observations on small graphs[18], the Birth-death process can also be a strong suppressor of selection and not just an amplifier. For example, detour graphs are a class of graphs that show strong suppression under the Birth-death process. These graphs are extremely assortative and the magnitude of the suppression is shown to depend on the detour size. We analytically find the optimal suppression sizes of the detour graphs for any given population size, and empirically show that the magnitude of maximum suppression decreases with population size. Since the lower bound of fixation probability decreases as population size increases, there are hints at the possible existence of an arbitrarily strong suppressor that neglects selective advantages in a large population. In biological settings, such as spatially complex ecosystems or cells in biological tissues, large populations are therefore more likely to be suppressors under the Birth-death update. If an event were to decrease the size of the population (for example, environmental catastrophes that lead to the destruction of forests or injury and aging of tissues), not only will the force of drift increase in the population, but also the suppressive capability of the population against the invasion of beneficial mutation will be compromised. One caveat is that this magnitude of suppression also depends on the strength of selection, and these topologies can transition from suppressor to amplifier given large enough selection pressure.

In rapidly cycling tissues, tissue maintenance and repair are coordinated by stem cells, which are routinely stochastically lost and replaced in the population[22] and thus instrumental for studying rates of evolution and mutation accumulation[23]. Previous theoretical studies on the population dynamics of stem cells either ignore the structure of the tissue, consider the topology of small populations of stem cells[55,56], or assume cells are arranged in lattices where every node has the same degree[57]. By analyzing recent imaging data on the spatial organization of hematopoietic stem cells in the bone marrow, we show that stem cell populations are organized to minimize fixation probabilities of new mutants spreading through the population.

While our focus here is to understand the evolutionary properties of the architecture of the stem cell niches, our approach makes many more questions ripe for exploration. For example, in a recent study, Watson et al.[58] use a well-mixed model to estimate mutation accumulation and selection coefficients in clonal hematopoiesis. Similarly, Heyde et al.[59] inferred division time and mutational fitness effects from variant allele frequency (VAF) data using a well-mixed Moran model, and found that increased stem cell proliferation expedites somatic evolution. Our results highlight that spatial heterogeneity can reduce the rate at which driver mutations spread through the population and suggest that using a well-mixed model to fit data produced by a spatially-structured population can potentially underestimate the strength of selection on somatic variants. Furthermore, in growing tumors, discrepancies can arise when sampling does not capture a uniform representation of the population, since over- or under-

representation of mutations in the VAF distributions due to spatial effects can be mistaken as signatures of selection[60]. The theory we present here can also be extended to study the evolutionary dynamics of spatially heterogeneous tumor populations[61–65].

Further work on how network properties shape evolutionary dynamics will also help us understand how to construct spatial structures in the limit of either suppression or amplification across various biological systems, natural or artificial. This would allow controlled suppression against the spread of unwanted variants and delay of population collapse. Reversely, we could also use population structure as a screening tool for faster amplification of newly discovered beneficial mutations or optimized protein complexes for medical or industrial applications.

## Methods
### List of network families used in the study
Linking network topology to evolutionary dynamics and understanding which network properties shape rates of evolution is complicated by the fact that these properties are often correlated, hard to tune independently and differ across many network families. In this study, we explore both well-known network families using built-in generators from NetworkX[66] and also design graphs that allow us to tune properties independently, as detailed below.

**Erdős Rényi random networks.** The Erdős Rényi model starts with a set of $N$ isolated nodes, and connects each pair of two nodes with probability $p$, independently. We generate Erdős Rényi random networks using built-in generators from NetworkX[66].

**Preferential attachment graphs.** In preferential attachment (PA) graphs, each network starts with a single node and nodes are added sequentially until the population reaches size $N$. Each new node is added to the network and connected to other individuals with a probability proportional to the individual's current degree to the power of a given parameter $\beta$. By adjusting the number of edges added each step ($m$) and the power of preferential attachment ($\beta$), this family of graphs allows for straightforward independent tuning of the moments of the degree distribution. Parameter $m$ is the only parameter of the model that controls the first moment of the degree distribution. Parameter $\beta$ controls the shape of the distribution, with the distribution being exponential when $\beta = 0$, stretched exponential when $0 < \beta < 1$, and power law when $\beta = 1$ (ref. 67). When the power of preferential attachment $\beta = 1$, PA graphs exhibit the scale-free property[28] and that is why PA graphs are often used as a model to study the spread of information or cultural norms[68].

**Random geometric graphs.** In contrast, for generalized random geometric graphs[31,32], nodes have spatial positions randomly drawn from a probability distribution to model spatially homogeneous populations (using the uniform distribution) or populations with heterogeneous spatial density (using the normal distribution). Once the spatial locations of the nodes are determined, the generating algorithm iterates through all pairs of nodes. An edge is created between two nodes using a probability distribution based on pair-wise distance. Here we use an exponential distribution (the resulting graphs are known as Waxman graphs) and a heavy-side function where we connect two nodes if the distance is below a predefined threshold (denoted as random geometric graphs).

**Small world networks.** We generate small world networks using built-in generators from NetworkX[66] and the Watts-Strogatz model.

**Detour graphs.** A detour graph is formed by starting with a complete graph of size $n_1$ and replacing one of the edges with a path of length $n_2 + 1 \geq 2$[43].

**Star graphs.** The star graph consists of one center node connected to $N-1$ outer nodes. We generate star graphs using built-in generators from NetworkX[66].

**Power-law cluster graphs.** We use the Holme and Kim algorithm for growing graphs with power-law degree distribution and approximate average clustering implemented in NetworkX[66].

**The graph rewiring algorithm**
We implement a sampling network generation algorithm based on simulated annealing (Fig. 1C) as follows: the algorithm relies on a degree swapping operation on graphs (ref. 21) and runs for a preset number of time steps. At every time step, two random edges are selected. Let us denote them by A–B and C–D. The two edges are broken and rewired to form A–C and B–D. The degree distribution of the graph is preserved, while other properties such as the mixing pattern of nodes are changed. Thus, we can use the edge swapping operation to find graphs with extreme graph properties from all possible graphs of fixed degree distribution. The algorithm can take any graph as input. We use degree Pearson correlation $r$ to measure the mixing pattern in the graph[27]. Parameter $r$ ranges from −1 to 1, with positive $r$ for networks where nodes with similar degrees are preferentially connected, and negative degree correlation for networks where high degree nodes preferentially form edges with low degree nodes.

If the goal is to find the graph that maximizes the degree correlation, we accept an edge swap according to the criterion:

$$\text{Uniform}[0,1] < \min\left(1, \exp^{-\gamma(r_{\text{after}}-r_{\text{before}})}\right) \tag{13}$$

and reject the step otherwise. Here, $1/\gamma$ is the annealing temperature that controls how stringent the criterion must be and is decreased as the simulation proceeds. Intermediate graphs are periodically saved and we use the heuristic outlined in Gkantsidis et al.[69] to periodically check that the graph is fully connected. This algorithm yields a set of graphs spanning a range of possible degree correlations, thus allowing us to study the effects of mixing pattern on evolutionary dynamics, without changes to the graph degree distribution.

We use combinations of parameters in five graph families and study six main graph characteristics (mean degree, variance in degree, third moment of degree distribution, network modularity, average clustering, and network assortativity), predictive of evolutionary dynamics. We use PCA to reduce the dimensionality of the data (Supplementary Fig. S3). We are able to capture 89% of the variation in network statistics using only the first three principle components. We observe that the graph families are clustered in this lower dimensional embedding space, with gaps in space existing between the network families. Therefore, our method allows us to sample graphs not accessible by traditional network generation algorithms specific to particular graph families. The black line in Supplementary Fig. S3 shows a continuous path through intermediate graphs, previously inaccessible using other methods, generated by the graph rewiring.

**Building the cellular networks of the stem cell niche architecture**
Every HSC niche constitutes a node in the graph and an edge is added between two nodes if the distance between them is less than a cut-off radius, similar to the generation of a random geometric graph. The samples vary in dimensions, number of cells, and segmentation techniques. We normalize the data by expressing the distance in units of the average distance between shortest pairs of cells (62.72 μm for HSC).

We use networks generated using cut-off distance of 15, which is the closest to our estimated biological interaction range. We interpret the cut-off distance as the maximum distance a HSC could travel in its entire lifespan. Live-animal tracking of individual hematopoietic stem cells in their niche showed MFG cells, a largely quiescent population with long-term self-renewal capability, displacing an average distance of 8.69 μm in a 2.5 h period[70]. HSCs have median replication time (the time when 50% of HSCs have divided) of 1.7 weeks[71]. During homeostasis, the rate of replication should balance the rate of depletion. This leads to the estimated interaction range of 1028 μm which corresponds to 16.4 × the distance between the shortest pairs.

In Fig. 6, we also show results for cut-off distances between 2 and 20, as gray dots, for comparison.

**Reporting summary**
Further information on research design is available in the Nature Portfolio Reporting Summary linked to this article.

## Data availability
Data required to reproduce analyses are available at https://github.com/yangpingkuo/Suppressor-of-selection-in-the-stem-cell-niches-of-the-bone-marrow. To build the networks of the stem cell niches of the bone marrow we use previously published open source datasets that provide the spatial location of hematopoietic stem and progenitor cells in four samples of mouse tibia[24] and the spatial locations of 8 bone marrow samples of CXCL12-abundant reticular cells, each with two images of two anatomically distinct regions, in total 16 cellular populations[25]. As stated in the studies referenced, all raw data are available upon request to the corresponding authors. Processed data derived from this study are available at https://github.com/yangpingkuo/Suppressor-of-selection-in-the-stem-cell-niches-of-the-bone-marrow.

## Code availability
Custom scripts were used for simulation studies and data analyses. Source code that allows for the reproduction of the simulations and results presented here can be found on Github at the following link: https://github.com/yangpingkuo/Suppressor-of-selection-in-the-stem-cell-niches-of-the-bone-marrow.

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

## Acknowledgements
We gratefully acknowledge support from the NIH National Institute of General Medical Sciences (award no. R35GM147445 to O.C.), the United States-Israel Binational Science Foundation (award no. 2019266 to O.C.) and from the NIH T32 training grant (no. T32 EB009403 to Y.P.K.). We also thank Daniel Coutu for help with the bone marrow imaging dataset used in the study. This research was done using computational resources provided by the Open Science Grid, which is supported by the National Science Foundation award 1148698, and the U.S. Department of Energy's Office of Science.

## Author contributions
Y.P.K. and O.C. conceived the study, performed the analyses and wrote the manuscript. C.N.A. contributed to the analyses involving the imaging data used in the study.

## Competing interests
The authors declare no competing interests.
