## [Peer Review File · Nature Communications]

nature portfolio

Peer Review File

A theory of evolutionary dynamics on any complex population structure reveals stem cell niche architecture as spatial suppressor of selectionREVIEWER COMMENTS

Reviewer #1 (Remarks to the Author):

The paper „Spatial architecture as a suppressor of selection in the stem cell niches of the bone marrow“ is a very interesting contribution, attempting to connect evolutionary graph theory to stem cell organisation.

Major comments:

- The paper is written in a style that I personally would judge as too pompous. For example, I do not think that this is the grand “unified evolutionary theory” that we have been waiting for. In line 83, it a general theory is announced, but I do not find this theory either very clear or very useful and general. However, I think this is an issue of presentation style (and often necessary to get into the review process) and should not cast doubt on the scientific content of the paper.
- The real innovation I see in the paper (which I really like) is the idea to keep track of the mutant frequencies for all nodes of the same degree only and thus greatly reduce the complexity of the issue. This is a very nice idea that can certainly help to take the field an important step further, but I do not see it as fully general, as it seems to work only for graphs that are sufficiently large and where nodes of the same degree are topologically similar. I am thus convinced that we can create a network where this approach does not work well. Please describe the possible pitfalls and assumptions of this approximation better!
- The derivation of the amplification factor α_B is nice and well done, but I think this has only relevance for graphs that are not well mixed. Thus, the purple lines in Fig. 3 are a bit misleading, I just see the well mixed case as a point for $\alpha_B=1$ that could be better shown as an arrow.
- Fig. 6 is the only direct connection to the title. I find it an interesting and thought-provoking application, but it is not clear which update mechanisms are at play and how this works in vivo. I would just tone down the title and sell this more as a possible future application of the framework.
- While the work on the hematopoietic system by Abkowitz et al. in 2000 was groundbreaking at its time, there is more recent work arguing for e.g. allometric scaling of the number of stem cells (Dingli & Pacheco, “Allometric Scaling of the Active Hematopoietic Stem Cell Pool across Mammals“, 2006) or great theoretical and experimental papers doing lineage tracing as in the wonderful work of Caroline Watson in Jamie Blundell’s group. I think these papers at the interface between experiment and theory deserve to be discussed. In the world of colonic crypts, there is lots of experimental work of the group of Hans Clevers and excellent models by Trevor Graham and his team which are also of interest, but of course not about the bone marrow.

Minor comments:

- Line 14: “Small regular networks” – what exactly do you mean with regular here? I do not think that e.g. the star graph is a regular network.
- Line 57: “Network approaches have mostly been used in population genetic theory to study evolutionary game-theory“ – I disagree with the wording here. I do not see a direct connection to population genetics here and evolutionary game theory is not an application of population genetics – on the contrary: This study is about constant fitness, which is more restricted than evolutionary game theory, but nonetheless this field is very interesting and rich. Imho, the vast majority of current studies of true games and graphs by now is rather boring and self-serving and have marginal relevance for the real world.
- In line 95, “we find that knowing the degree distribution alone is surprisingly not enough to

determine the fixation probabilities and times“- I do not think that the authors are seriously surprised about this, people working in this field (including the senior author) do know this quite well!

- Moeller 2019 was published together with a paper by Tkadlec et al. in the same journal, I suggest to cite them jointly, as they are closely related.

- Line 148: “speed up or suppress adaptation” – please be more precise if you mean probabilities or times.

- Line 303: There are amplifiers of selection stronger than the star, see e.g. Pavliogannis et al., “Amplification on Undirected Population Structures: Comets Beat Stars” (2017).

- Fig. 1B is a very beautiful illustration and the rewiring trajectory is interesting to see. However, Fig. 1C would be easier to understand if the lower right 3-node had three links attached to it and if the links going “outward” would be a bit longer.

- In Fig 2, it would be better if the legend about approximations would be in a panel that actually includes the associated approximation.

- In the SI captions please spell out how the “previous approximation” has been made and provide a relevant citation.

Reviewer #2 (Remarks to the Author):

Authors present a Moran model w/ biallelic genotypes with fitness of 1 or $1+s$, with birth-death or death-birth update schema, and derive some analytical results that describe the role of spatial structure on amplifying or suppressing selection. In general, the topic is very timely, and the authors clearly have a good grasp of the relevant literature, and give a good exposition of the fundamental results in the field of the graph structure and selection amplification.

The manuscript presents the results very cleanly and thoroughly in the figures, although some of the captions and within text explanation of the figures can be more detailed to avoid confusion. At the same time, the overall length of the manuscript is long, especially in relation to typical publications in this venue, and I think it can be substantially shortened (especially the introduction and discussion) without impacting the novelty / reach of the paper.

Major comments:

1. I am confused by the statement in line 318, that degree distribution alone is not sufficient to determine fixation probability, when fixation probability is accurately predicted by distribution mean/variance in figure 2. What are the limits of this approximation, does it only work for PA graphs?

2. Comments on Figure 2:

a. It's not intuitive to me why the lines in Figures 2C and D are varying length. If the degree distribution is a function of the graph/network, should not the range of variation be the same for all the lines.

b. Figure 2: is it possible to derive an analytical approximation for Time to Fixation for Bd and dB processes, given the approximation derived for Fixation Probability? These approximations are noted in the legend, but not found in panels B, D.

c. I think it would also be reasonable to run Monte Carlo simulations of well-mixed graphs, showing the agreement w/ the pink analytical approximation.

3. Comments on Figure 4:

a. The caption is not sufficiently explained – please elaborate on what panels A, B, C

represent.

b. In the main text, it appears that panel C are graphs w/ identical degree distribution. Are these the same graphs as A? It's not clear to me how these same graphs can collapse onto one axis when panel A shows a high degree of noise and panel B shows a non-constant function of alpha as a function of r, and panel C is presumably not a function of r.

4. Comments on Figure 6:

a. The connection to real-world data is appreciated. A common research question in HSC is to infer the fitness effect (e.g. "s" parameter in your model) of observed mutations based on variant allelic frequency data. It would be more interesting to repeat the analysis in figure 6 with varying values of fitness, to show the likelihood of observing mutations as a function of fitness and population size. This final figure is a nice contribution, but I feel that it doesn't answer a very interesting biological question, and should be revisited before publication. For example, Moran models (Heyde et al., 2021, Cell) are used to infer fitness of mutants in HSC populations. How can the models presented here improve fitness inferences? It may be also useful to look at Watson et al., Science 367, 1449–1454 (2020), and other publications using similar methods.

5. I believe a comment can be made (within the discussion, perhaps) on the application of this theory to growing populations (e.g. cancer), if some progress can be made in this direction.

Minor comments:

Line 72: "small perturbation" – perturbations of what? Perturbing the network structure, I presume?

Line 137: "edge pattern imposes limits to variant spread" – what does this mean? I am not sure the authors consider edge patterns, so perhaps best to remove.

Line 190: "degree Pearson correlation r" – in general, the correlation parameter here is under-explained as to why the authors are using this particular metric.

Figure 3: I defer to the journal's preference for such things, but I suggest the authors consider using the same Y-axis limits for both panels A and B, for ease of comparison.

Figure 4: in x-label in panel B, add "r" to the label, since "r" is referenced in the main text.

Line 335: there is a type "[b]f Figure 4c"

Figure 5: What do the inset graphs on the top right represent? Please label/annotate.

Thank you for considering our paper and allowing us to respond to reviews! We have worked to reply to all the reviewer comments and we are very grateful to the reviewers for their time and thoughtful suggestions.

Briefly, the main changes made as a direct reply to the reviewers' comments and suggestions are these (but see detailed descriptions throughout this document):

1. We have tightened up and rewrote entire sections of the introduction and discussion, at the suggestion of both reviewers. We have included a discussion of assumptions and pitfalls as suggested by reviewer 1. We have also changed the title as suggested by reviewer 1, to include the main theoretical theme of the paper but kept the reference to the main result of the application of our theory since it is an important finding that needs to be highlighted.

2. We have added a new panel to the application figure (Figure 6) to show the robustness of our results relative to the process used (birth death or death birth). The result shows the suppression of fitness is consistent regardless of Bd or dB update. We also changed the selection strength from 0.005 to 0.01, corresponding to 1% fitness increase. This is to address reviewer 2's comment on mutation fitness estimation from VAF data since most literature is framed in terms of the relative fitness increase of the mutations.

3. We show the results for different cutoffs under the death-birth process in Supplementary Figure S6. We have also added Supplementary Figure S7 in response to reviewer 2's comment on repeating the analysis in Figure 6 for different fitness values.

4. We have made changes throughout the manuscript and figure captions to improve the flow of writing and better highlight our results.

REVIEWER COMMENTS

Reviewer #1 (Remarks to the Author):

The paper „Spatial architecture as a suppressor of selection in the stem cell niches of the bone marrow“ is a very interesting contribution, attempting to connect evolutionary graph theory to stem cell organisation.

We thank the reviewer for the kind comments and suggestions: we have worked to address them all and believe the paper is much improved as a result.

Major comments:

- The paper is written in a style that I personally would judge as too pompous. For example, I do not think that this is the grand “unified evolutionary theory” that we have been waiting for. In line 83, it a general theory is announced, but I do not find this theory either very clear or very useful and general. However, I think this is an issue of presentation style (and often necessary to get into the review process) and should not cast doubt on the scientific content of the paper.

We have worked to rewrite some sections of the paper, in particular the introduction and discussion sections to tighten them up and address the reviewer’s comments.

We would argue that being able to extend analytic results to a much wider array of heterogeneous graphs and spatial structures and present how we can unify graph properties into an evolutionary parameter of amplification and suppression, as well as showing ability to build networks that span this parameter from some of the strongest amplifier to the strongest suppressors, in our opinion, does constitute a general theory that allows us to organize, design and understand.

However, obviously, one can design exceptions to every rule and approximation. This does not diminish their value.

We did take the reviewer’s comments seriously and hopefully the writing of the paper has greatly improved.

- The real innovation I see in the paper (which I really like) is the idea to keep track of the mutant frequencies for all nodes of the same degree only and thus greatly reduce the complexity of the issue. This is a very nice idea that can certainly help to take the field an important step further, but I do not see it as fully general, as it seems to work only for graphs that are sufficiently large and where nodes of the same degree are topologically similar. I am thus convinced that we can create a network where this approach does not work well. Please describe the possible pitfalls and assumptions of this approximation better!

We agree that this type of approach makes certain assumptions that do not apply to all possible networks, but nonetheless its value lies in the fact that it allows us to understand and tackle evolutionary dynamics on a much wider array of networks than before, by a large degree. The way we think of it is that very small networks can be tackled using incidence matrix approaches, and they can be very exact since the incidence matrix uniquely determines the network. This approach however does not work on larger networks, and this is the gap we fill here. The degree distribution-based approach we take here allows us to greatly expand our understanding of evolutionary dynamics on networks, even though the degree distribution, as opposed to the incidence matrix, does not uniquely define the network structure. We think this is a reasonable trade-off since the full model is incredibly hard to solve for a large system.

We have added additional discussion of this and the limitations of the approach in the Discussion section, lines 451-456. For example, our approach would not work when a graph consists of two parts connected by a few edges, where one part is highly assortative, and one part is highly disassortative. Since our approach averages over the entire graph, we would treat the graph as neutrally assortative. This would lead to incorrect prediction of the fixation probability.

We also note that the approach ignores higher-order organizations of the network, such as community structures and network motifs.

- The derivation of the amplification factor α_{dB} is nice and well done, but I think this has only relevance for graphs that are not well mixed. Thus, the purple lines in Fig. 3 are a bit misleading, I just see the well mixed case as a point for $\alpha_{dB}=1$ that could be better shown as an arrow.

We agree that for figure 3, the well-mixed case corresponds to a dot at $\alpha = 1$. We have used the line to represent the well-mixed case since the role of the spatial structure is really highlighted by the difference with the dynamics in the well-mixed populations. We have changed Figure 3 to show well-mixed as a point with dashed lines pointing to it for comparison to other graphs.

- Fig. 6 is the only direct connection to the title. I find it an interesting and thought-provoking application, but it is not clear which update mechanisms are at play and how this works in vivo. I would just tone down the title and sell this more as a possible future application of the framework.

We agree Fig. 6 in its present form seems to suggest Birth-death update is the underlying mechanism in the HSC population dynamics. However, it is important to note that the results hold for both the Birth-death and death-Birth updates. The results for both update processes strongly suggest the bone marrow spatial structure to be suppressor of selection. This is important (and we have made it clearer to the reader) since this means the results do not depend on the update process assumed, which is not the case for a lot of other spatial structures. We chose to not present the death-Birth plot originally, since most graphs are suppressors under death-birth update, and the surprising result is with the Bd update.

For emphasis of the result, we have now changed Figure 6 and have added a panel with the dB update, to show that the analysis is agnostic to update mechanism.

We also take the reviewers point that our theoretical results are more general than the application and link to data we choose to explore in the current manuscript. We have added the more theoretical topic of the paper but also chose to keep the main result of the application as we believe it is important and needs highlighted to prospective readers.

- While the work on the hematopoietic system by Abkowitz et al. in 2000 was groundbreaking at its time, there is more recent work arguing for e.g. allometric scaling of the number of stem cells (Dingli & Pacheco, "Allometric Scaling of the Active Hematopoietic Stem Cell Pool across Mammals", 2006) or great theoretical and experimental papers doing lineage tracing as in the wonderful work of Caroline Watson in Jamie Blundell's group. I think these papers at the interface between experiment and theory deserve to be discussed. In the world of colonic crypts, there is lots of experimental work of the group of Hans Clevers and excellent models by Trevor Graham and his team which are also of interest, but of course not about the bone marrow.

Thank you for pointing us these results and citations! We have now reshaped the discussion section and discuss the significance of our results in the context of these references and potential future directions. Specifically, in addition to Watson, 2021, already cited on line 482, we also include these references in the new discussion paragraphs on lines 480-492.

These works seem to suggest that there is no consensus on the number of active HSC at any given time: Dingli, 2006 suggests 111-385 active HSC for humans, while Watson, 2020 suggests upwards to 25000. Lee-Six, 2018 seems to suggest more, 50,000-200,000. For mouse, the number of HSC would be smaller, but we think our results of 200-4000 HSC fall within the reasonable range. The allometric scaling of the hematopoietic system (Dingli, 2006) strengthens our results, since we observe more suppression at larger population sizes. We have added a discussion of this on lines 415-422.

Minor comments:

- Line 14: "Small regular networks" – what exactly do you mean with regular here? I do not think that e.g. the star graph is a regular network.

Thank you for pointing out a potentially confusing phrasing. The sentence is trying to convey that most previous theory is done at two extremes: 1) wide range of small networks and 2) large but regular networks (lattices) or very particular patterns (stars).

We have changed line 14 to "However, they usually assume either very small networks, large but regular networks, or with strong constraints on the strength of selection considered."

We have also changed (new) line 58 from 'on very small, regular graphs' to 'on very small graphs'.

- Line 57: "Network approaches have mostly been used in population genetic theory to study evolutionary game-theory" – I disagree with the wording here. I do not see a direct connection to population genetics here and evolutionary game theory is not an

application of population genetics – on the contrary: This study is about constant fitness, which is more restricted than evolutionary game theory, but nonetheless this field is very interesting and rich. Imho, the vast majority of current studies of true games and graphs by now is rather boring and self-serving and have marginal relevance for the real world.

We take the reviewer's point and have removed the sentence the sentence and now cite the relevant work together with others on line 56 as part of a more general comment.

- In line 95, "we find that knowing the degree distribution alone is surprisingly not enough to determine the fixation probabilities and times"- I do not think that the authors are seriously surprised about this, people working in this field (including the senior author) do know this quite well!

We have removed surprisingly from the introduction, line 95.

- Moeller 2019 was published together with a paper by Tkadlec et al. in the same journal, I suggest to cite them jointly, as they are closely related.

We have added the citation to Tkadlec et al, 2017 on lines 289 and 330.

- Line 148: "speed up or suppress adaptation" – please be more precise if you mean probabilities or times.

We have changed the sentence to the phrase "speed up or suppress adaptation through shaping probabilities and times to fixation of new mutants in the population" (new line 129).

- Line 303: There are amplifiers of selection stronger than the star, see e.g. Pavliogannis et al., "Amplification on Undirected Population Structures: Comets Beat Stars" (2017).

Thank you for pointing to us to comet graphs. We have changed line 289 to "one of the strongest amplifiers for undirected graphs" and added the reference to Pavliogannis, 2017. We have also changed line 329 to "star graphs, known to be one of the strongest undirected amplifiers."

- Fig. 1B is a very beautiful illustration and the rewiring trajectory is interesting to see. However, Fig. 1C would be easier to understand if the lower right 3-node had three links attached to it and if the links going "outward" would be a bit longer.

Thank you for pointing to the missing link on the lower right 3-node. We corrected figure 1C by adding the missing edge, and we made all outgoing edge longer and more visible.

- In Fig 2, it would be better if the legend about approximations would be in a panel that actually includes the associated approximation.

We have moved the legends for figure 2, from 2B to 2A and from 2D to 2C, to reflect the panels with the approximation lines.

- In the SI captions please spell out how the “previous approximation” has been made and provide a relevant citation.

For the previous approximation we have used analytical results for weak selection from (“Fixation probabilities in evolutionary dynamics under weak selection” by Mcavoy, 2021). We have added the relevant citations in the caption of figures S1 and S2.

Reviewer #2 (Remarks to the Author):

Authors present a Moran model w/ biallelic genotypes with fitness of 1 or $1+s$, with birth-death or death-birth update schema, and derive some analytical results that describe the role of spatial structure on amplifying or suppressing selection. In general, the topic is very timely, and the authors clearly have a good grasp of the relevant literature, and give a good exposition of the fundamental results in the field of the graph structure and selection amplification.

The manuscript presents the results very cleanly and thoroughly in the figures, although some of the captions and within text explanation of the figures can be more detailed to avoid confusion. At the same time, the overall length of the manuscript is long, especially in relation to typical publications in this venue, and I think it can be substantially shortened (especially the introduction and discussion) without impacting the novelty / reach of the paper.

Thank you for taking the time to review our paper. We really appreciate the comments and suggestions and for helping us improve our paper and the exposition of the results.

We have worked to provide more detailed explanations throughout the results section (and especially around the figure captions and descriptions) to increase the clarity of our results and ease of understanding for a reader.

We have also worked to significantly tighten up and shorten the introduction and discussion sections and to keep only the necessary references and paragraphs.

Major comments:

1. I am confused by the statement in line 318, that degree distribution alone is not

sufficient to determine fixation probability, when fixation probability is accurately predicted by distribution mean/variance in figure 2. What are the limits of this approximation, does it only work for PA graphs?

Thank you for pointing out how our phrasing there can be confusing. As we discuss in the section starting on page 10 “The evolutionary role of the graph mixing pattern”, while for the dB process the degree distribution is enough for prediction, for the Birth-death process the mixing pattern (edge connection properties, not just the properties of nodes) also matters. And we show that the amplification parameter for the Bd process that is presented in Figure 3B also incorporates the mixing pattern. That paragraph is meant to be a connection between these results in Figure 3B and Figure 4, where we dive deeper into this dependency and actually show the independent effect of the mixing pattern on the value of the amplification parameter for a graph (as we keep mean and variance in degree fixed), as well as on the probability of fixation.

We now clarify this in the text and the sentence /this transition now reads: “Since knowing the degree distribution alone is not enough to determine the fixation probability and amplification parameter for the Birth-death process, to illustrate the effects of assortativity and mixing pattern on fixation probabilities without the influence of degree distribution and graph generating method, we use edge swap operations to sample graphs with different mixing patterns, while keeping the degree distribution the same.” (line 303)

The approximation for the Bd process depends on the degree distribution and mixing pattern and in figure 2 we plot over variance on the x axis. The variance in degree correlates with mixing pattern and we continue to disentangle the two network parameters and their roles further in figure 4.

2. Comments on Figure 2:

a. It's not intuitive to me why the lines in Figures 2C and D are varying length. If the degree distribution is a function of the graph/network, should not the range of variation be the same for all the lines.

The lines in Figures 2C and D (showcasing the role of variance in degree) have different lengths because the range of variance for graphs we can test with simulations depends on what mean degree we fix. Every line of a different color contains simulations for graphs of the same mean degree, and the smaller the mean degree the harder it is to tune graphs to have very high variance. The analytic lines do not have this issue, but we decided to have them also cover the parameter range of the simulations to avoid confusion.

It is really a combinatoric problem to tune graphs of fixed mean degree and varying variance in degree: how many distinct graphs can you have for a certain number of edges in the network? There is only one graph with size 100 that has mean degree 99 (complete graph) with variance 0. The number of graphs with size 100 that has mean

degree 98 is significantly larger. Therefore, the variance can change more for different graphs with the same mean.

b. Figure 2: is it possible to derive an analytical approximation for Time to Fixation for Bd and dB processes, given the approximation derived for Fixation Probability? These approximations are noted in the legend, but not found in panels B, D.

While it is possible to use the analytic approaches presented here to derive approximations for the time to fixation, our focus in this paper is on getting an intuitive unified understanding of the probability of fixation, so our analysis on time is limited. In addition we have another manuscript currently under revision that focuses on the problem of time to fixation and how it is controlled by higher-order network motifs.

As the reviewer notes, the confusion stems from us placing the legends for all Panels in Panels B and D. We have moved the legends in 2B to 2A and in 2D to 2C, to make it clear that we only have the approximation for probabilities of fixation.

We have also added additional detail in the Figure 2 caption to say which equation is used for the approximations in Figure 2 A and C and that we use methods outlined in Ewens 2004 for the well-mixed approximation lines, present in all panels.

c. I think it would also be reasonable to run Monte Carlo simulations of well-mixed graphs, showing the agreement w/ the pink analytical approximation.

Thank you for this comment. While it is very important to have a strong knowledge of the baseline well-mixed model to compare our results against, the reason we didn't include simulations for it is because the the well-mixed model is so very well studied, and the theory so well known and known to match simulations results very well. Any additional ones we feel would make the already busy plots additionally so.

We now include and highlight the relevant citations (see lines 238-239) to previous results from the well-mixed model to make it easier for the reader to find the relevant comparison literature.

3. Comments on Figure 4:

a. The caption is not sufficiently explained – please elaborate on what panels A, B, C represent.

We have added additional details in the caption of Figure 4 to better explain the results presented in panels A B and C.

b. In the main text, it appears that panel C are graphs w/ identical degree distribution. Are these the same graphs as A? It's not clear to me how these same graphs can collapse onto one axis when panel A shows a high degree of noise and panel B shows a

non-constant function of alpha as a function of r, and panel C is presumably not a function of r.

For figure 4, the graphs presented in Panels B and C are same as the graphs in Panel A. The reviewer is right that the graphs in A show a large range of fixation probabilities for a given mean and variance in degree. This is what we wanted to show, that the mixing patterns or correlation in degree r (the pattern of how the edges are distributed across nodes of different degrees) also plays a large role in shaping the probabilities of fixation, for the Bd process.

A common measure of mixing pattern is the degree correlation coefficient, r . Panel 4B shows that the variance in P_{fix} we see in 4A is indeed due to variation in r . But this dependence on r is not the same across graph types (the non-constant function part that the reviewer noticed in the graph types). This has to do with the degree distribution we mentioned in the previous section. Here r simply measures mixing pattern and does not take degree distribution into account. This is why alpha, the graph amplification factor presented in 4B and C is a better metric for predicting evolutionary dynamics (and specifically probabilities of fixation), because it takes both degree distribution and mixing pattern into account.

We have added additional detail in the text and the figure caption to make the results clearer to the reader.

4. Comments on Figure 6:

a. The connection to real-world data is appreciated. A common research question in HSC is to infer the fitness effect (e.g. “ s ” parameter in your model) of observed mutations based on variant allelic frequency data. It would be more interesting to repeat the analysis in figure 6 with varying values of fitness, to show the likelihood of observing mutations as a function of fitness and population size. This final figure is a nice contribution, but I feel that it doesn't answer a very interesting biological question, and should be revisited before publication. For example, Moran models (Heyde et al., 2021, Cell) are used to infer fitness of mutants in HSC populations. How can the models presented here improve fitness inferences? It may be also useful to look at Watson et al., Science 367, 1449–1454 (2020), and other publications using similar methods.

Thank you for the pointing us to the work by Watson, 2020 and Heyde 2021 (which we now cite). We agree that inferring fitness of mutants in HSC populations is an extremely interesting application, and we are currently working on a larger follow-up project towards this goal.

For example, attached below is a result we obtain for running simulations to produce VAFs using the bone marrow network we build here. It showcases that the spatial structure does indeed affect generated variant allele frequencies, compared to the well-mixed model. The well-mixed Moran model developed in the papers mentioned above can fit the VAF for data generated from the spatially structured bone marrow architecture, but would predict a different selection coefficient than the ground truth

(around 10% lower). As a result, the fitness inferred using well-mixed model on VAF from space will lead to biased fitness estimations.

The exploration of these differences, how they depend on the parameters and network structures and how to correct for these biases is a full project in itself and beyond the scope of the current paper. Adding an entire section explaining VAFs would add to an already long paper.

We agree with the reviewer that repeating figure 6 with varying values of fitness, to show the likelihood of observing mutations as a function of fitness and population size is a nice addition. We now include this in the Supplementary Material Figure S7. We also change s from 0.005 to 0.01 (to represent 1% fitness advantage) in Figure 6. We added a figure similar to Figure 6 but with fitness 0.05, and 0.1, corresponding to fitness advantage of 5% and 10%, as Supplementary Figure 7

We have also added a new panel to Figure 6 to showcase the death-birth process next to the birth-death process and show that the results hold regardless of process used.

5. I believe a comment can be made (within the discussion, perhaps) on the application of this theory to growing populations (e.g. cancer), if some progress can be made in this direction.

We have shortened and rewrote the discussion section and now have a paragraph on further applications of the theory (including the spatially heterogeneous evolving ecosystem that is cancer): lines 480-492.

Minor comments:

Line 72: “small perturbation” – perturbations of what? Perturbing the network structure, I presume?

Indeed, by perturbation we mean perturbation to the network structure, away from very regular structures. We have now changed the wording in (former) line 72 to read: “small perturbations to the network structure”, now on line 66.

Line 137: “edge pattern imposes limits to variant spread” – what does this mean? I am not sure the authors consider edge patterns, so perhaps best to remove.

We have changed the statement to: “individuals reproduce locally, and their offspring spread to neighboring nodes connected by an edge.” (line 117)

Line 190: “degree Pearson correlation r ” – in general, the correlation parameter here is under-explained as to why the authors are using this particular metric.

We have added clarification and added a citation to Newman, 2002 for details to what the degree Pearson correlation measures. In addition, we have also added explanations to what different values of r mean in terms of how we should interpret its effect on network structure in line 172: “Parameter r ranges from -1 to 1, with positive r for networks where nodes with similar degrees are preferentially connected, and negative degree correlation for networks where high degree nodes preferentially form edges with low degree nodes”.

Figure 3: I defer to the journal’s preference for such things, but I suggest the authors consider using the same Y-axis limits for both panels A and B, for ease of comparison.

Thank you. We argue for keeping the Y-axis different for panels A and B and the main reason is that using the same Y-axis for both figures wouldn’t allow the fuller, more zoomed-in view of all the graphs for the dB process and better identification of all the color differences. We think it’s best to zoom-in and not leave unused white space, so readers can easily see the difference between graph families.

Figure 4: in x-label in panel B, add “ r ” to the label, since “ r ” is referenced in the main text.

We have added “r” to the label in figure 4B.

Line 335: there is a typo “]bf Figure 4c”

We apologize for the latex typo, we have corrected “]bf Figure 4c” to “\bf Figure 4c” in new line 321.

Figure 5: What do the inset graphs on the top right represent? Please label/annotate.

Thank you for bringing this to our attention: the inset graphs on the top right represent detour graphs and we now specify this in the caption of Figure 5.

REVIEWERS' COMMENTS

Reviewer #1 (Remarks to the Author):

In my opinion, the paper has been greatly improved in the revision and I no longer have the impression that this is too pompous or exaggerated. The authors convincingly replied to all my comments, thank you for taking them very serious. I still have a few minor issues listed below (only some of them may trigger a slight modification), but I would leave it to the authors to decide if they wish to implement any change.

- Line 102: „We use the structure of a graph to represent the spatial structure of the population“. This is a detail, but I feel that you convincingly argue that graphs capture also topological structures that go beyond “spatial”. I constantly struggle with the notion of space we use in the field, but in the end it is a matter of taste what to use.
- Line 186: Here it would be good to mention the different initialization schemes, for Bd it has been argued that temperature initialization is more appropriate, but this of course depends on the process generating mutations.
- Line 203: “...even for $N = 23$ nodes it can take several minutes“ – The paper cited for this is already quite old and the algorithm may not be well designed. I think it may be better to argue that already for N smaller than 30, this approach becomes unfeasible.
- Line 224: “the solution to the partial differential equation“ – to me, this still is a PDE, despite an approximated one. The solution is to me only Eq.6. There, it may also help to mention that the dB solution is the same with different coefficients.
- Line 288: „one of the strongest amplifiers for undirected graphs“ – the wording is fine, but so far, this is known only for small graphs. It will be interesting to see one day if this also holds for larger graphs!
- SI: References seem to be broken.
- SI, Eq. 12: I find it a bit unlucky to replace variables by Landau Symbols within an equation. But this may be a convention in other fields and the area is very interdisciplinary. Similarly, I could not expand Eq. 17 in this notation, as in my field $O(y)$ is arbitrary and does not contain any further information.

Reviewer #2 (Remarks to the Author):

I thank the authors for addressing my previous comments, as I believe all my previous concerns were addressed in a satisfactory manner. I had some misunderstandings when reading the original version, and the new exposition of the results is very much improved and clarified. My only remaining concern is the title. I do agree some inclusion of a general result is warranted (as suggested by reviewer 1), but I am unsure what 'adaptive properties' as I believe the only appearance of the word adaptive is in the title. Perhaps the original title is fine, without the mention of the application: "spatial architecture as a suppressor of selection" -- yet I leave this to the authors' & editor's preference.

Reviewer #1 (Remarks to the Author):

In my opinion, the paper has been greatly improved in the revision and I no longer have the impression that this is too pompous or exaggerated. The authors convincingly replied to all my comments, thank you for taking them very serious. I still have a few minor issues listed below (only some of them may trigger a slight modification), but I would leave it to the authors to decide if they wish to implement any change.

- Line 102: „We use the structure of a graph to represent the spatial structure of the population“. This is a detail, but I feel that you convincingly argue that graphs capture also topological structures that go beyond “spatial”. I constantly struggle with the notion of space we use in the field, but in the end it is a matter of taste what to use.

We agree and have now changed the text to read: “We use the structure of a graph to represent the structure of reproduction and replacement of the population.”

- Line 186: Here it would be good to mention the different initialization schemes, for Bd it has been argued that temperature initialization is more appropriate, but this of course depends on the process generating mutations.

We mention the different schemes directly after this: “In the first update scenario, we assume reproduction occurs before death (the Birth-death Bd scenario).”

- Line 203: “...even for $N = 23$ nodes it can take several minutes“ – The paper cited for this is already quite old and the algorithm may not be well designed. I think it may be better to argue that already for N smaller than 30, this approach becomes unfeasible.

We have made this change: “(even for $N=23$ nodes, the approach becomes unfeasible).”

- Line 224: “the solution to the partial differential equation“ – to me, this still is a PDE, despite an approximated one. The solution is to me only Eq.6. There, it may also help to mention that the dB solution is the same with different coefficients.

That section now says:” the solution can be approximated using....” to clearly highlight how we use the following equation to approximate the solution.

- Line 288: „one of the strongest amplifiers for undirected graphs“ – the wording is fine, but so far, this is known only for small graphs. It will be interesting to see one day if this also holds for larger graphs!

We have made this change, adding the known (so far): “An example of this is the star network, one of the strongest known amplifiers for undirected graphs.”

- SI: References seem to be broken.

We have fixed the latex compilation error.

- SI, Eq. 12: I find it a bit unlucky to replace variables by Landau Symbols within an equation. But this may be a convention in other fields and the area is very interdisciplinary. Similarly, I

could not expand Eq. 17 in this notation, as in my field $O(y)$ is arbitrary and does not contain any further information.

We agree, however we believe the current notations help us tighten up the equations and clarify the results for the reader.

Reviewer #2 (Remarks to the Author):

I thank the authors for addressing my previous comments, as I believe all my previous concerns were addressed in a satisfactory manner. I had some misunderstandings when reading the original version, and the new exposition of the results is very much improved and clarified. My only remaining concern is the title. I do agree some inclusion of a general result is warranted (as suggested by reviewer 1), but I am unsure what 'adaptive properties' as I believe the only appearance of the word adaptive is in the title. Perhaps the original title is fine, without the mention of the application: "spatial architecture as a suppressor of selection" -- yet I leave this to the authors' & editor's preference.

We agree with the reviewers that the unifying theoretical advances of our paper are currently not part of the paper title and we have attempted to come up with a new version that mentions both the theoretical advances of the paper and the application:

“A theory of evolutionary dynamics on any complex population structure: stem cell niche architecture as a spatial suppressor of selection”